

# Lagrangian Coherent Structures to Examine Mixing in the Stratosphere

Jezabel Curbelo[1,2,3] and Marianna Linz[4, 5]

[1]Universitat Politécnica de Catalunya. Departament de Matemàtiques. Barcelona. Spain
[2]IMTech. Institute of Mathematics of UPC-BarcelonaTech, Barcelona, Spain
[3]Centre de Recerca Matemàtica, Bellaterra, Spain
[4]Harvard University School of Engineering and Applied Sciences
[5]Harvard University Department of Earth and Planetary Sciences

**Correspondence:** Jezabel Curbelo (jezabel.curbelo@upc.edu)

**Abstract.** The study of mixing in the stratosphere is important for understanding the transport of chemical species and the dynamics of the atmosphere. How best to quantify this mixing is not settled, however. In recent years, Lagrangian Coherent Structures (LCSs) have emerged as a valuable tool for examining mixing in fluid flows, and in this work we present a stratospheric mixing metric based on the LCS framework. We identify LCSs associated with the transport of air masses and

quantify the amount of mixing between different regions of the atmosphere in the Whole Atmosphere Community Climate Model (WACCM). Our results show that LCSs provide a powerful approach to analyze mixing in the stratosphere and can be used to identify regions of high and low mixing as well as to study the dynamics of the atmosphere. The results are compared with those obtained two other tools to quantify mixing: the commonly used effective diffusivity and the recently introduced isentropic eddy diffusivity. We find qualitative agreement between these metrics for much of the stratosphere, although there

are regions where they clearly disagree. A significant advantage of the LCS mixing metric is that it reflects Lagrangian transport in physical latitude rather than the equivalent latitude coordinate needed to calculate effective diffusivity, and we discuss other advantages and disadvantages of these methods.

## 1 Introduction

The stratospheric circulation shapes the distribution of trace gases in the stratosphere, including ozone, water vapor, and

ozone depleting substances (Butchart, 2014). This circulation can conceptually be separated into two components: the slow meridional overturning circulation and the fast quasi-horizontal mixing. These are not independent—the breaking of planetary-scale waves drives both—but the timescales are sufficiently different that the separation is physically meaningful. Metrics that quantify the stratospheric circulation typically employ this separation by timescale. The meridional overturning circulation is often characterized with the vertical velocity in the transformed Eulerian mean (TEM) circulation (Andrews and McIntyre,

1976) or less frequently with the diabatic circulation (Pyle and Rogers, 1980; Rosenfield et al., 1987; Linz et al., 2017), which is very similar to the TEM vertical velocity calculated from radiative heating rates (Linz et al., 2019). Methods for calculating the quasi-horizontal mixing have less consensus, with mixing being characterized by a Lagrangian effective diffusivity (Nakamura,



1996; Haynes and Shuckburgh, 2000; Abalos et al., 2016a) or with a box model (Neu and Plumb, 1999; Ray et al., 2010, 2016),
using Lyapunov diffusivity (Shuckburgh et al., 2009; d'Ovidio et al., 2009), Lagrangian diffusivity (Curbelo and Mechoso,
2024; LaCasce, 2008), vertical gradients of the ideal age tracer (Linz et al., 2021; Gupta et al., 2023) or even as a residual in
the age tracer budget (Garny et al., 2014; Ploeger et al., 2015a, b).

   The balance between the slow overturning and the fast quasi-horizontal mixing determines the distributions of trace gases.
For example, lower-stratospheric mid-latitude ozone is affected by mixing of ozone poor air from the tropics and downward
transport of ozone rich air from above (e.g., Hegglin and Shepherd (2007); Neu et al. (2014)). This lowermost stratospheric
region is important for air quality and for climate; it is the reservoir of stratospheric air from which air is transported into the
troposphere via stratospheric intrusions (Shapiro, 1980; Holton et al., 1995), and ozone is a particularly strong greenhouse
gas in this region. Analysis of tracers in models shows that they have relatively similar representations of the strength of the
overturning circulation but very different mixing strength (Dietmüller et al., 2018). Likewise, although future ozone changes
due to advection are somewhat consistent in models, the ozone changes due to differences in eddy mixing changes are much
more uncertain both because of the processes and because of the methods to calculate mixing terms (Abalos et al., 2020).

   Chaotic advection describes a situation in which a spatially smooth and periodic or quasiperiodic time-dependent velocity
field produces irregular particle trajectories Aref (1984); Ottino (1989). This definition can be generalized in a natural way to
temporally aperiodic kinematic flows such as stratospheric flows Malhotra and Wiggins (1998): chaotic mixing and transport
are mediated by the lobes formed by the stable and unstable manifolds Wiggins (1992) and explained using hyperbolicity
concepts associated with attracting and repelling lines, as well as hyperbolic zones with strongly chaotic behavior. Therefore,
a necessary condition for chaotic advection in dynamical flows is that there exists an "organizing structure" (that includes
stagnation points) for mixing and transport Ngan and Shepherd (1999). This framework for studying fluid transport and mixing
is often referred to as the 'dynamical systems approach to Lagrangian transport' since the focus is on understanding the
"organizing structures" in phase space for fluid particle trajectories (Ottino and Wiggins (2004); Samelson and Wiggins (2006);
García-Garrido et al. (2018))

   There is strong evidence for the existence of such spatial structure in the stratosphere. Waugh and Plumb (1994) and Norton
(1994) showed that small-scale tracer structure in the stratosphere is determined mostly by the large-scale flow. This behav-
ior is consistent with the so-called Rossby wave critical-layer paradigm (e.g., Juckes and McIntyre (1987), which states that
the forced Rossby wave critical layer provides a useful conceptual model for the stratospheric surf zone (Haynes and McIn-
tyre (1987); Salby and Garcia (1987); Bowman (1996)). Several authors (e.g., Bowman (1993); Waugh and Plumb (1994);
Schoeberl and Newman (1995)) have shown that particle trajectories in the surf zone are chaotic.

   According to Ottino (1989), fluid mixing can be conceptualized as the effective stretching and folding of material lines and
surfaces. Even turbulent flows feature Lagrangian structures that can be understood from the perspective of finite-time dynam-
ical systems theory Shadden et al. (2005). This has led to the development of the theory of Lagrangian Coherent Structures
(LCS) Haller (2015). A LCS is a distinguished surface within a dynamical system, like fluid flow, that significantly influences
the behavior of nearby trajectories over a specific time interval. LCS play a crucial role in shaping global transport and act
as transport barriers by serving as key material surfaces. These finite-time structures are the most relevant structures in time-





dependent flows, as they determine the deformation of the fluid and the evolution of any advective tracer field. The concept of LCSs was introduced by Haller and Yuan (2000). Attracting hyperbolic LCSs are lines that evolve with the flow and attract
fluid to the greatest extent. In the vicinity of these attracting LCSs, the fluid is stretched in one direction and compressed in the other direction, making them the cores of filamentous tracer patterns. The application of LCSs to the stratosphere has the potential to provide new insights into the mechanisms controlling the transport and mixing of chemical species.

This paper introduces the density of Lagrangian Coherent Structures as a new metric for stratospheric mixing and compares it to two different stratospheric mixing metrics: the more commonly used effective diffusivity Nakamura (1996) and the newer
tracer-based isentropic eddy diffusivity Gupta et al. (2023). Although effective diffusivity is commonly used, its calculation is computationally expensive and results are in the somewhat ambiguous equivalent latitude coordinate. As we will show in section 3.2.1, latitude and equivalent latitude are not (name withstanding) truly equivalent. The density of LCS does not solve the problem of computational expense, but it is a Lagrangian metric that nevertheless provides results in physical latitude.

We also compare to the isentropic eddy diffusivity, a metric based on the eddy transport of the idealized tracer age of air
and the mean meridional gradient in age of air Gupta et al. (2023). Age of air describes how long an air parcel has been in the stratosphere since entering at the tropopause, and so age reflects the average of the Lagrangian pathways of the bits of air that make up a larger air parcel. At each location, there is more accurately an age spectrum that describes the distribution of times different parts of the air parcel have taken to reach the current point (Hall and Plumb, 1994). Mean ages in the stratosphere reach over 5 years, and so the computational burden of using age is due to model equilibration time. Age of air, more than any
other tracer, reflects Lagrangian transport. The isentropic eddy diffusivity is calculated from age transport, and is implicitly a Lagrangian metric. We introduce the density of LCS as a mixing metric not with the suggestion that it is superior to these other methods but with the idea that it provides a different and hopefully useful perspective on stratospheric mixing.

The paper is organized as follows: In Section 2, we describe the data and methods used in our analysis. In Section 3, we apply the proposed metric (section 2.2) to characterize mixing and present our results, while also comparing its use to the other
metrics defined in Section 2.3 and 2.4. We present our conclusions and discuss both limitations and potential avenues for future research in Section 4.

## 2   Methods

### 2.1   Model simulation

The study utilized a comprehensive climate model, the Community Earth System Model 1 Whole Atmosphere Community
Climate Model (WACCM), which is identical to the one used in Linz et al. (2021) and Gupta et al. (2023). This interactive chemistry-climate model (Garcia et al., 2017; Marsh et al., 2013) was developed at the National Center for Atmospheric Research (NCAR) and incorporates physical parameterizations to simulate complex earth system processes, such as atmospheric chemistry and radiation. It is based on a finite-volume dynamical core (Lin, 2004) from the Community Atmosphere Model, version 4 and covers a domain from the surface to 140 km, with 31 pressure levels (Neale et al., 2013). The model has a hori-
zontal resolution of 2.5° longitude × 1.875° latitude, corresponding to the F19 horizontal grid. The WACCM simulations were



based on the Chemistry Climate Model Initiative REF-C1 scenario (Morgenstern et al., 2017) and forced with observed sea surface temperatures. The Quasi-Biennial Oscillation was nudged to observed winds, but otherwise, the model evolved freely. The model was integrated from 1979, and additional information on WACCM is provided in Section 3 of Linz et al. (2021).

## 2.2 Lagrangian Descriptor

The approach is based on the Lagrangian descriptor called the $M$ function (Mancho et al., 2013). By analyzing trajectories in both the forward and backward directions, this technique can categorize trajectories with analogous qualitative characteristics. The expression for the $M$ function is as follows:

$$M(\mathbf{x}_0, t_0, \tau) = \int_{t_0-\tau}^{t_0+\tau} \|\mathbf{v}(\mathbf{x}(t; \mathbf{x_0}), t)\| dt, \tag{1}$$

and corresponds to the arc length of the trajectory traced by a fluid parcel starting at $x_0 = x(t_0)$ as it evolves forwards and
backwards in time with velocity $\mathbf{v}(\mathbf{x}, t)$ for a time interval $(t_0 - \tau, t_0 + \tau)$. This tool already has been widely used to visualise LCS in atmospheric flows (de la Cámara et al., 2012; Manney and Lawrence, 2016; Curbelo et al., 2017, 2019a, b, 2021; Niang et al., 2020).

LCSs are often used to study mixing in fluid flows because they can be used to identify regions of the flow where there is a high degree of mixing and regions where there is little mixing. In particular, regions where LCSs are closely packed together
or highly curved are associated with strong mixing, while regions where LCSs are far apart or relatively straight are associated with weak mixing.

Figure 1 shows snapshot of the euclidean norm of the horizontal gradient of M, $\|\nabla M\|$, at 600K for several days of 2008. The curves on isentropic surfaces where $\|\nabla M\|$ has large magnitudes (in black) approximate LCSs (Curbelo et al. (2019a, 2021)). By considering that a higher quantity of LCS correlates with increased mixing, we can infer from Figure 1 that the midlatitudes
have greater mixing than the equator and tropics, with variations also apparent due to seasonal differences.

To measure mixing based on the notion that higher spatial occurrence of LCS leads to greater mixing, we have developed a diagnostic tool that relies on the probability density function (PDF) of LCS presence across each time for a given latitude, longitude and isentropic level.

The algorithm works as follows: for each instant (daily) and isentropic level of the wind velocity, we calculate $M$ using
equation (1) and then determine the LCS as the ridges of the gradient of $M$, $\|\nabla M\|$, following Curbelo et al. (2019a, 2021), i.e., the locations of LCS in maps of $M$ correspond to abrupt spatial changes in descriptor, which are identified by large values of normalized $\|\nabla M\|$. We then calculate the PDF in the corresponding time interval for each specified latitude, longitude, and altitude. Finally, we compute the zonal average of this quantity, which is shown in the upper row of Figure 2 for the climatological period 2007-2009 (panel a), for December - January- February 2007-2009 (DJF, b) and June-July-August 2008-
2009 (JJA, c). This map displays the locations where coherent structures are most prevalent (in yellow), indicating higher levels of diffusion, in contrast to areas where their presence is lower (dark blue).



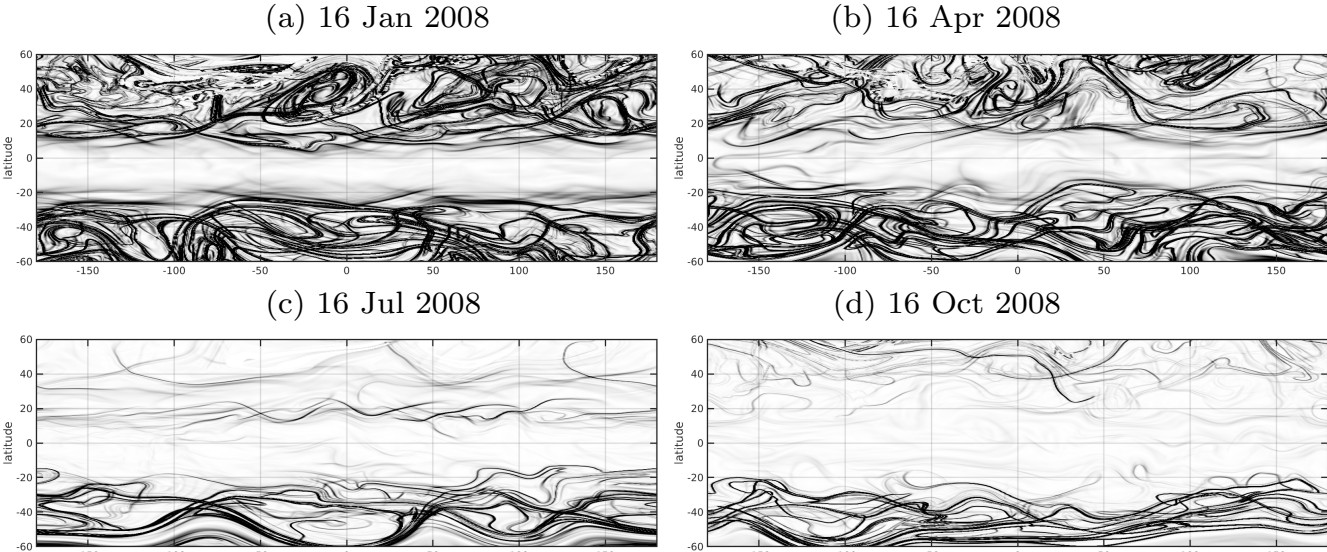

**Figure 1.** Snapshot of $\|\nabla M\|$ for different days of the year 2008 at 600K. In the maps, black lines correspond to large values of $\|\nabla M\|$, that is, approximately the LCSs.

Note that it is not necessary to take the zonal mean. LCS-density can be used to look at the full three-dimensional structure of stratospheric mixing (cf. Shuckburgh et al. (2009)). For comparison with other metrics of mixing, however, the zonal mean is the most appropriate.

Previous studies have explored the spatial structure of mixing using both dynamical and tracer-based metrics. We compare our new diagnostic tool based with another Lagrangian tool, effective diffusivity, and the implicitly Lagrangian isentropic eddy diffusivity from age.

## 2.3    Effective diffusivity

The effective diffusivity represents the degree of intricacy or elongation of tracer contours as they are transported by the
non-divergent isentropic wind field. This method, which is based on a Lagrangian treatment of mixing and transformation of equations based on tracer-area coordinates, has been used by Nakamura (1996); Shuckburgh et al. (2001); Abalos et al. (2016a) among others to quantify mixing in the atmosphere. Following Haynes and Shuckburgh (2000), the effective diffusivity $\kappa_{eff}$ in equivalent latitudes $\phi_e$ is defined by:

$$\kappa_{eff}(\phi_e, t) = \kappa \frac{L_{eq}^2(\phi_e, t)}{(2\pi R \cos \phi_e)^2} \tag{2}$$

where $L_{eq}$ is the equivalent length of a tracer contour, the diffusivity $\kappa$ is a constant and $R$ the Earth's radius. The equivalent latitude $\phi_e$ is defined by $A = 2\pi R^2 (1 - \sin \phi_e)$, being $A$ the area of the region for which the tracer concentration is greater than or equal to the tracer contour $Q$. By using equivalent latitude as an independent variable, the advective terms in the tracer





concentration evolution equation are eliminated. This means that the effective diffusivity $\kappa_{eff}$ only characterizes the tracer's diffusion with respect to the contours of $\phi_e$.

The second row of Figure 2 shows the effective diffusivity in equivalent latitude coordinates $\phi_e$ versus $\theta$ coordinates. To compute (2) we use as a tracer the potential vorticity (PV) and we follow the calculations of Haynes and Shuckburgh (2000). Regions characterized by high values of effective diffusivity (in yellow) demonstrate strong mixing, causing tracer contours to be extensively stretched.

### 2.4    Isentropic eddy diffusivity

An alternative approach to evaluate the meridional distribution of the mixing flux is through the isentropic eddy diffusivity. Eddy tracer fluxes have been used in the past to examine mixing (e.g., Plumb and Mahlman, 1987; Abalos et al., 2016b), and the version of this calculation we think is most relevant for comparison to our LCS metric is the one used in Gupta et al. (2023) because it is based on age of air. Age of air, $\Gamma$ is a measure of how long an air parcel has been in the stratosphere, and age has a spatially independent source (of 1 year/year) within the stratosphere with a boundary condition of zero at the tropopause

Hall and Plumb (1994); Waugh and Hall (2002). Age represents the integrated paths of the bits of air that make up a larger parcel, and so we expect age to be more closely related to the Lagrangian perspective than other tracers. The isentropic eddy diffusivity $\mathcal{D}_{eff}$ (units m$^2$ s$^{-1}$), calculated from age and following the notation of Gupta et al. (2023), is the the ratio of the net eddy transport of age to the mean meridional age gradient and is defined by:

$$\mathcal{D}_{eff} = \frac{1}{\overline{\rho_\theta}} \frac{-F_{eddy}}{\partial_y \tilde{\Gamma}} \tag{3}$$

being $\tilde{\Gamma}(\phi,\theta) = \overline{\rho_\theta \Gamma}/\overline{\rho_\theta}$ the mass weighted age in isentropic coordinates, $\rho_\theta$ the isentropic density, $v$ the meridional velocity, $\partial_y = (1/R)\partial_\phi$ the meridional gradient of the mean isentropic age and $F_{eddy}$ the eddy age flux in isentropic coordinates given by

$$F_{eddy}(\phi,\theta) = \overline{\rho_\theta v \Gamma} - \overline{(\rho_\theta v)}\tilde{\Gamma}. \tag{4}$$

Here the overbar denotes zonal averaging on fixed isentropes. The eddy diffusivity given by $\mathcal{D}_{eff}$ is expected to have its
highest value in the midlatitudes of the stratosphere due to the strong eddy transport induced by planetary waves and the weak meridional gradients in the surf zone. Consequently, the isentropic eddy diffusivity can be utilized as a qualitative measure to evaluate the midlatitude eddy mixing structure. The isentropic eddy diffusivity defined by (3) for the same period as previous tools is shown in the lowermost row of Figure 2.

## 3    Results

### 3.1    Relation between the stratospheric circulation and the presence of Lagrangian Coherent Structures

To better understand the results of using the new LCS metric for mixing, first we briefly review some characteristics of the stratospheric circulation, its seasonal cycle, and what might be expected for the seasonal cycle of stratospheric mixing. Much





**Figure 2.** Probability density function of the presence of LCS (first row), Effective diffusity $\kappa_{eff}(\phi_e, \theta)$ distribution (second row) and Isentropic eddy diffusivity $\mathcal{D}_{eff}(\phi, \theta)$ (third row), averaged over each day during the years 2007-2009. Columns: (a) Climatology 2007 - 2009 (b) DJF 2007-2009 (c) JJA 2007-2009. The dashed black line is the zero diabatic velocity curve, i.e. $\dot{\theta} = 0$. This line is a function of $\phi$ and $\theta$.

of this explanation is based on Plumb (2002) and Butchart (2014), and we highlight the characteristics most relevant to understanding mixing. The large-scale stratopsheric circulation is driven by the breaking of planetary scale Rossby waves that propagate upwards from the troposphere. These waves can be caused by land-sea contrast, flow over topography, or the interactions of synoptic-scale eddies. Waves can propagate vertically only in appropriate background winds, and the appropriateness of the winds is dependent on the zonal wavenumber (Charney and Drazin, 1961). The summer hemisphere has mean easterly





winds above the lower part of the stratosphere prohibiting deep vertical propagation of Rossby waves. We thus expect waves to break in the lower stratosphere in the summer hemisphere and the upper part of the stratosphere to be relatively quiescent. The

winter hemisphere, in contrast, has climatological westerly winds that are typically of a strength that allows zonal wavenumbers 1-2 to propagate vertically into the middle and upper stratosphere (waves with zonal wavenumbers 3-4 still break in the lower stratosphere). If we consider that wave breaking is associated with horizontal mixing, we thus expect to see strong mixing in the lower stratosphere in both hemispheres and mixing in the winter hemisphere throughout the depth of the stratosphere. Other features that are important are the subtropical mixing barriers (the so called tropical pipe (Plumb, 1996; Neu and Plumb,

1999)) and the polar vortex. The polar vortex is a band of strong westerly winds at about 60 degrees latitude in the winter hemisphere, and these winds are well known as a mixing barrier (e.g., Mitchell et al. (2021)).

The new metric based on the presence of LCS is calculated for the period of 2007-2009, and the annual average and the solstice seasons are shown in the top row of Figure 2. Here, the dashed black lines represent the turnaround latitude in each hemisphere, that is, the latitude associated with zero diabatic velocity $\dot{\theta} = 0$. The turnaround latitude can have a varying latitude

with height and time. The region bounded by the two dashed black lines, known as the tropical pipe, exhibits a gradual diabatic ascent of mass. Similarly, the area located toward the poles, encompassed by the black curves, is characterized by a slow diabatic descent of mass. These two divisions are referred to as the upwelling and downwelling regions, respectively. In DJF, the Northern Hemisphere has a greater stronger circulation and the upwelling region extends into the Southern Hemisphere. The maximum in LCS-density is poleward of the diabatic circulation minimum, but equatorward of the vortex edge. This suggests

the greatest mixing is occurring in the surf zone between these two barriers. Similarly, in JJA, the Southern Hemisphere has a stronger circulation and the upwelling area is extended into the Northern Hemisphere. The with the highest values of LCS-density around 30S, just poleward of the southern turnaround latitude.

Figure 2 shows regions of low LCS-density include the regions at the core of the polar vortices, with lowest values in the SH polar jet in JJA around 60S. There is a gradient from the highest values of LCS-density in the midlatitudes to lower

values near the subtropical mixing barriers. The lowest LCS-density is then found at the equator, suggesting very little mixing happening within the deep tropics (5°S-5°N) in either season. This is consistent with the picture shown in the snapshots in Figure 1, where there are almost no strong gradients in M right at the equator. Our findings for the tropics are consistent with the effective diffusivity results from Abalos et al. (2016a) (more below) who point out that this is to be expected because of known weak mixing within the tropical pipe Plumb (1996).

**3.2 Comparison between density of LCSs and effective diffusivity**

A comparison of the three metrics used to describe diffusion and mixing, effective diffusivity, LCS-density and isentropic eddy diffusivity, is shown in Figure 2. Our discussion of the features somewhat modeled on that in Abalos et al. (2016a), whose presentation of effective diffusivity is excellent.

In the region above the subtropical jets, the most intense mixing occurs in the middle latitudes of the lower stratosphere

during the summer. This mixing pattern extends to higher altitudes during the austral (December-February) summer than the





the boreal (June-August) summer. In the winter lower stratosphere, the mixing is constrained to a narrower latitudinal band due to the presence of the lower portion of the polar vortex. This is especially true for the Southern Hemisphere.

The evolution as a function of latitude for the effective diffusivity and the LCS-density is shown in Figure 3 on four selected isentropic levels: 400 (the lower part of the stratospheric overworld region), 450, 500 and 800K (lower, middle and high

stratosphere). For consistency with the calculation of LCS where we use a time interval $\tau = 10$ days, here we also apply a 10-day running-mean to daily diffusivity. Both metrics show characteristics in common: minimum values are observed in the core of the polar jets in both hemispheres and in the tropics, following the latitudinal displacement of the tropical easterlies towards the summer hemisphere. In the middle stratosphere, mixing gradually increases at the edge of the vortex during late winter and spring, persisting until approximately one month after the breakdown.

Differences arise in the seasonal evolution in the northern hemisphere. Effective diffusivity has a near-constant, high value at 450 K around 40-50 N, while the LCS shows weaker mixing in summertime (as expected based on wave breaking). In contrast, the seasonal cycle for effective diffusivity is stronger higher up, with stronger summertime mixing at 800 K.

There is also a difference in relative magnitudes of apparent mixing lower and higher in the stratosphere, with LCS showing a stronger gradient between 400 K and 800 K. This could be real and reflect a difference in wave breaking that is better

captured with the LCS, or it could be related to the choice of a constant $\tau$ at all levels. A lower $\tau$ is appropriate for looking at tropospheric variability, for example. All levels here are within the stratosphere, but we perform no optimization for choosing $\tau$, and so this choice could introduce a bias.

Figure 4 highlights the temporal evolution of the diffusivity for equivalent latitude of 50N on the same isentropic levels as figure 3. This emphasizes that the seasonality very present when using the LCS-density in the middle stratosphere is lost

in the case of using the effective diffusivity (see the yellow and purple curves corresponding to 500K, 800K respectively). A strong seasonal cycle in mixing is expected because of the seasonality of wave breaking. Thus, the lack of a strong seasonal cycle in effective diffusivity suggests the metric may be lacking. We hypothesized that this may be related to the seasonality in equivalent latitude, which we now investigate.

### 3.2.1 Equivalent latitude

The utilization of equivalent latitude, derived from contours of potential vorticity (PV), has become a widely accepted method for analyzing the transport of air masses on isentropic surfaces within the stratosphere and upper troposphere (Allen and Nakamura, 2001, 2003; Allen et al., 2012; Nakamura, 2024). However, employing equivalent latitude based on potential vorticity (PV) as a horizontal coordinate has some limitations (Allen and Nakamura, 2003; Pan et al., 2012). Firstly, the calculation of PV involves intricate mathematical manipulations of observed or assimilated winds and temperature, resulting in

the introduction of noise due to the convolution of analysis errors (Newman, 1989). Although this noise does not significantly affect the equivalent latitude overall, it can be amplified at specific locations and times when PV gradients are weak. Secondly, the resolution of PV is constrained by the resolution of the underlying wind and temperature fields, whose data quality decreases with altitude, making PV notably noisier in the upper stratosphere (Manney et al., 1996). To address these problems, several





**Figure 3.** Diffusivity distribution using our new diagnostics tool (LCS-density, first column) and effective diffusivity (second column) for January 2007 to July 2009 on the 400K, 450K, 500K and 800K isentropic surfaces. A 10-day running-mean has been applied to the daily diffusivity.





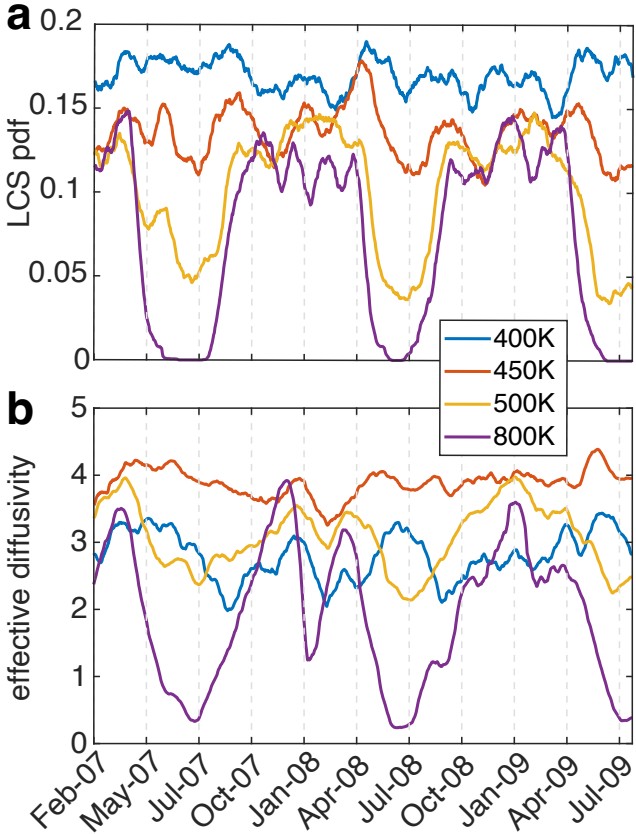

**Figure 4.** a) Distribution of LCS at $\phi = 50N$ and b) Effective diffusivity distribution at $\phi_e = 50N$ for January 2007 to July 2009 on the 400K, 450K, 500K and 800K isentropic surfaces. A 10-day running-mean has been applied to the daily diffusivity.

authors have developed alternative methods for generating high-resolution maps of equivalent latitude for tracer analysis (Allen and Nakamura, 2003; Haynes and Shuckburgh, 2000; Anel et al., 2013).

However, the utilization of equivalent latitude coordinates also presents other inherent complications. Equivalent latitude and latitude may not be exactly equivalent (despite the name), as we now show. Therefore the effective diffusivity mixing results in equivalent latitude may not be directly comparable to mixing metrics in latitude. In order to understand the effect of the use of coordinate systems based on equivalent latitude in the definition of diffusivity, we examine equivalent latitude: its temporal evolution, mean and standard deviation.

The standard deviation of the equivalent latitude for each isentropic level and PV contour values $Q$ defined such as

$$\sigma(Q,\theta) = \sqrt{\frac{1}{N-1}\sum_{i=1}^{N}|\phi_e^i(Q,\theta) - \overline{\phi_e}(Q,\theta)|}, \tag{5}$$





**Figure 5.** The equivalent latitude $\phi_e$ as a function of time $t$ and PV contour $Q$ $(\phi_e(Q,t))$ for the isentropic level 400, 450, 500 and 800K. Last panel shows the time evolution of $\phi_e$ for $Q = -2 \cdot 10^{-5} m^2 s^{-1} K kg^{-1}$ on these $\theta$−levels.

where $N$ is the number of observation (days), $\phi_e^i(Q,\theta)$ is the corresponding latitude value of the area enclosed by the closed contour $Q$ on the level $\theta$ at time $t_i$, and $\overline{\phi_e}(Q,\theta)$ is the time mean:

$$\overline{\phi_e}(Q,\theta) = \frac{1}{N} \sum_{i=1}^{N} \phi_e^i(Q,\theta).$$ (6)

To simplify the notation, we relate the $Q$ contour that depends on the isentropic level with the temporal mean of the equivalent latitude on that surface, that is, we take $\phi_e \equiv \overline{\phi_e}(Q,\theta)$ and therefore, $\sigma(\phi_e,\theta) \equiv \sigma(Q,\theta)$.





**Figure 6.** JJA Season: (a) Standard deviation of the equivalent latitude (in degrees latitude) and (b) effective diffusivity as a function of the equivalent latitude and isentropic levels. (c) probability density function of the presence of LCS and (d) isentropic eddy difusivity as a function of the latitude and isentropic levels.

Figure 5 shows the temporal evolution of the equivalent latitude for each potential vorticity contour value at four different potential temperature levels $\theta = 400, 450, 500$ and $800K$. The last panel of this figure illustrates the changes in the equivalent latitude, $\phi_e$, associated with a given value of potential vorticity, $Q$, across different isentropic levels and over time. It reveals regions where $\phi_e$ remains nearly constant (near the equator), in contrast to other regions where significant differences are observed. Low latitudes show little variability, while higher latitudes show seasonal and interannual variability. The strong seasonal cycle in the PV contour associated with a given equivalent latitude suggests that the seasonal cycle in effective





diffusivity could be related to the seasonal evolution of the coordinate system. Effectively, by using equivalent latitude, we are
imposing the seasonal cycle of PV onto the mixing metric.

To examine this effect further, Figure 6(a) shows the standard deviation of the equivalent latitude following this definition
(5) for the JJA season 2007-2009. The standard deviation is compared with effective diffusivity (panel (b)), density of LCS (c)
and isentropic eddy diffusivity (d). Large values of standard deviation of equivalent latitude coincide with the regions where
the greatest differences between the three metrics are found, i.e., in the South Hemisphere between 50-20S and in the Northern
Hemisphere at lower isentropic levels and inside the band of 40-20N for higher isentropic levels. This last region is particularly
interesting because, despite the fact that $\kappa_{eff}$ shows large values of diffusivity here, the other metrics, which do not use $\phi_e$,
show much lower diffusivity values (LCS-density) or even indicate non-existent diffusivity in the area as in the case of $\mathcal{D}_{eff}$.
This is a region where relatively few waves can propagate and break, and so low values are more consistent with the wave
forcing.
Regarding the Southern hemisphere, in the places where $\sigma$ is smaller (60-40S), we have a better correspondence between
$\kappa_{eff}$ and LCS-density. For higher $\sigma$-values, the differences between this two metrics are more noticeable, particularly in
magnitude. These similarities and differences related to the standard deviation of the equivalent latitude qualitatively support
our idea that merely the fact that effective diffusivity is necessarily calculated in $\phi_e$-coordinates can lead to different results.
(We note this is a cautionary example of treating equivalent latitude and latitude as the same quantity.)

### 3.3 Comparison of LCS-density with isentropic eddy diffusivity

The isentropic eddy diffusivity is presented in the bottom row of Figure 2, and we see that it generally agrees rather well
with the overall structures for both effective diffusivity and LCS-density. Some noticeable differences are present, including
the higher values of isentropic eddy diffusivity in the tropics and the apparent lack of any subtropical mixing in the summer
hemisphere. The higher tropical values are consistent with prior results (Abalos et al., 2016b) comparing a tracer-based eddy
flux diffusivity to effective diffusivity. The subtropical difference is not as evident in the previous results, however. As effective
diffusivity and the LCS-density are both lower in the deep tropics, especially as the effective diffusivity should have no effects
of time variation of equivalent latitude at these low latitudes, it seems likely that the isentropic eddy diffusivity is providing an
overestimate of the mixing. In the lower stratosphere, the LCS-density likely overestimates the diffusivity due to the election
of constant $\tau$ in $\theta$ (as mentioned above). However, effective diffusivity is also likely over-estimated between 400-500K in the
NH due to the differences between equivalent latitudes in that area (see panel a of Figure 6). Here, isentropic eddy diffusivity
likely gives us the most reliable picture below 500K, despite the fact that there is better agreement between the LCS-density
and effective diffusivity.

## 4 Summary and Discussion

In this paper we develop a new diagnostic method to quantify mixing from wind velocities on isentropic surfaces following a
Lagrangian approach. The proposed method is based on the definition of Lagrangian Coherent Structures (Haller and Yuan,





2000), which are special features of fluid flow that are closely linked to the mixing of passive tracers. Using a comprehensive climate model, we present a new and effective approach for quantifying mixing in the atmosphere without relying on equivalent latitude coordinates. This quantitative method establishes a direct relationship between LCS characteristics and mixing/diffusivity measures.

The results obtained using this technique exhibit strong consistency with previous studies, highlighting regions of pronounced mixing near the boundaries of the polar vortices and subtropical edges. Furthermore, areas characterized by limited mixing are identified within the central portion of the vigorous westerlies and within the tropical pipe. Specifically, we have compared the density of LCS to effective diffusivity (Nakamura, 1996) and isentropic eddy diffusivity (Gupta et al., 2023).

We find that the use of the equivalent latitude coordinate likely aliases the seasonal cycle of PV with the seasonal cycle of
mixing by examining the variability in equivalent latitude with time. This effect is larger in some regions than others, with little variation present in the tropics, and far more in the subtropics and midlatitudes. We therefore caution that using the latitude-equivalent coordinate ($\phi_e$) may mask certain details of the seasonality.

In summary, our LCS-based metric offers a robust diagnostic tool for comprehending the intricate dynamics of fluid mixing and transport. Furthermore, they can be effectively employed to quantify mixing within the lower and middle stratosphere
through the utilization of output from reanalysis products. The LCS-based measures can also provide estimates of the meridional span of adiabatic mixing in the winter/summer midlatitudes and serve as a framework to establish connections between diabatic upwelling and adiabatic mixing, shedding light on the underlying differences between these processes. The LCS-density metric has some important limitations, however.

Like isentropic eddy diffusivity, effective diffusivity, and aging by mixing, the LCS-based mixing we examine here cannot
be calculated directly from observations. The mixing metrics that can be calculated from observations are based on the TLP model (Ray et al., 2016; Linz et al., 2021), and so they are representing only a mixing coefficient at each latitude rather than a more comprehensive picture of mixing. A good method for calculating spatially resolved diffusivity directly from observations has yet to be developed, as far as the authors are aware.

While the new method discussed offers valuable insights into mixing regions, it does not fully address the challenge of
computational cost due to the expense associated with trajectory calculations. By relying solely on wind velocity, it remains particularly useful for preliminary modeling studies, model output, and initial testing phases. We acknowledge its limitations concerning the selection of temporal integration time intervals, especially as the density of LCS depends on the integration time and the temporal scale likely varies with altitude, being different in the stratosphere and troposphere. In the future, we aim to conduct a more in-depth investigation into this topic.

In addition, the LCS-density is a measure of mixing more akin to "aging by mixing" (Garny et al., 2014) than to a true diffusivity, as it is not formulated to have the appropriate units to serve as a coefficient for diffusion. This is something that was addressed for a Lagrangian structures-based metric using Lyapunov diffusivity (d'Ovidio et al., 2009), but that metric has coefficients that were fit to best match effective diffusivity and is calculated in equivalent latitude. We contend that the qualitative reasoning behind the link between the LCS-density and eddy mixing is sound, but a theoretical relationship between the
LCS-density and diffusivity would enable a more useful diffusion metric. Unlike the limitations of lack of a ready comparison



with data and of the expensive computational needs, this theoretical formulation as a diffusivity is something that could be addressed in future work.

We have introduced an LCS-based mixing metric that is a true Lagrangian metric that nevertheless avoids equivalent latitude. Overall, it agrees quite well with two other metrics for mixing, and we have discussed the disagreements. The applicability

of this metric is currently limited because it cannot be calculated directly from observations, because it needs high temporal resolution model output, and because it is not directly a diffusivity estimate. There is hope for theoretical progress that would enable a more rigorous definition for a diffusivity based on LCS density and we also intend to explore some of the vertical dependence of the integration time in the future.

*Code and data availability.* The data used come from the Community Earth System Model 1 Whole Atmosphere Community Climate Model

(WACCM), which is identical to the one used in Linz et al. (2021) and Gupta et al. (2023) and are available here: https://doi.org/10.7910/DVN/GBRCWW (Linz et al., 2021). The age of the air data product is the same as that used in Linz et al. (2017) and is available here: https://doi.org/10.6084/m9.figshare.5229844.v1.

*Author contributions.* Both authors, JC and ML, designed the study. JC performed the calculations and made the figures. Both authors, JC and ML, discussed the results and wrote the paper.

*Competing interests.* The authors have no competing interests to declare.

*Acknowledgements.* We acknowledge Douglas E. Kinnison for providing the high temporal resolution WACCM output, including high temporal resolution spatially-resolved Age of Air data. JC also acknowledges the support of the RyC project RYC2018-025169, the Spanish grant PID2020-114043GB-I00, PID2021-122954NB-I00 and CNS2023-144360, and the "2022 Leonardo Grant for Researchers and Cultural Creators, BBVA Foundation". ML was funded by NASA awards 80NSSC21K0943 and 80NSSC23K1005.



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
