# Peer review of "Lagrangian Coherent Structures to Examine Mixing in the Stratosphere"

_EGUsphere, 2024_

## Referee Comment (RC1)

Review of manuscript egusphere-2024-1348, with title "Lagrangian Coherent Structures to examine mixing in the stratosphere", by Jezabel Curbelo and Marianna Linz.

This paper introduces a metric based on the density of singular features of a Lagrangian descriptor to diagnose mixing in the stratosphere. The new metric is compared with existing mixing diagnostics in a zonal mean (or equivalent latitude) perspective using the output of the climate model WACCM, and the results show good qualitative agreement over all. The manuscript is well written and the results are potentially interesting, but some extra analysis and clarifications are needed. After these, it should be suitable for publication in ACP.

Main comments:

1) Although stated in several parts of the text (lines 76-77, 314), it is not clear "what new perspective the new method provides on stratospheric mixing", since the comparison with other mixing diagnostics gives an overall agreement.
I think the paper does not explore one obvious strength of the LCS-density diagnostic: the fact that it can diagnose large-scale stirring in longitude-latitude. The paper would greatly benefit from analyzing maps of LCS-density and relate the diagnostic with the main features of the stratospheric circulation, such as the location of the polar vortices, jets, regions of wave breaking, etc.

2) About section 3.2.1. If I understand correctly, the idea is that since equivalent latitude ($\phi_{eq}$) has a seasonal cycle in PV-coordinates, then the seasonal cycle of effective diffusivity ($\kappa_{eff}$) is affected by this feature and might be "contaminating" the physical significance of $\kappa_{eff}$.

Latitude and equivalent latitude are not equivalent indeed, as stated in lines 242-243. The former is a geographical coordinate, the latter is a tracer contour-based coordinate: it represents the latitude at the edge of a polar cap enclosing the same area as a given tracer contour. Or in other words, it represents the area enclosed by a given tracer contour. Given a "well-behaved" tracer, at a given time there should be a one-to-one correspondence between a tracer contour value and $\phi_{eq}$. So if I am not mistaken, plotting the time-evolving PV as a function of $\phi_{eq}$ should be equivalent to plotting the time-evolving $\phi_{eq}$ as a function of PV (as in Fig. 5), it is a matter of remapping one as a function of the other. All this to say that the seasonal cycle of $\phi_{eq}$ in PV-coordinates should be the same as the seasonal cycle of PV in $\phi_{eq}$-coordinates. In fact, since the zonal-mean PV is monotic in latitude, one could perform a similar exercise and obtain a variable called "latitude" as a function of a "zonal-mean PV-coordinate"; one would expect an equivalent seasonal cycle in PVzonalmean(lat) and in lat(PVzonalmean).
The point I am trying to make is that $\kappa_{eff}$, as long as it is computed using PV as a tracer, will indeed be affected by the seasonal cycle of PV (lines 257-260), but this should be independent of $\kappa_{eff}$ being defined in equivalent latitude coordinates, since PV has a seasonality in any coordinate.
[By the way, it is surprising that the negative contour of PV at 800K in the last panel of Fig. 5 is placed at $\phi_{eq}$~40ºN, it does not look consistent with the fourth panel of Fig. 5 where the $\phi_{eq}$=40ºN-contour is located at positive PV values.]

3) This takes me to my next comment. That band of high $\kappa_{eff}$ at $\phi_{eq}$ = 20º-40ºN in JJA (Fig. 6b) is indeed quite interesting. As reflected in the text, there should be no dynamical reason for high mixing to occur in that region since Rossby waves are filtered out below in the summer stratosphere. In fact, that band does not appear in the reanalysis ERA-Interim (see Fig. 1c in Abalos et al. 2016). However, the $\kappa_{eff}$ calculations in Abalos et al. were performed by integrating a diffusion-advection model to evolve a passive tracer, as conceived by Nakamura (1996). So the question remains whether that band appears when using PV to calculate $\kappa_{eff}$.

I happen to have calculated $\kappa_{eff}$ for the same WACCM runs (CCMI refc1) that are used in the present study, as well as for ERA-Interim, that I used in a study some years ago. I have plotted in Fig. R1 a

similar plot as Fig. 6b, but for a 60-year mean in WACCM (Fig. R1a), and for a 34-year mean in ERA-Interim (Fig. R1b). I also get that high $\kappa_{eff}$ over 20º-40ºN in WACCM in a 60-year average, but that region is not present in the reanalysis, either computing $\kappa_{eff}$ with PV or with a passive tracer (see Fig. 1c in Abalos et al. 2016). This suggests that the high $\kappa_{eff}$ in the summer stratosphere is not a characteristic feature of $\kappa_{eff}$ computed with PV, but it is true that it does show up in WACCM. Moreover, Fig. R1a (WACCM) also displays a region with very high $\kappa_{eff}$ right inside the austral polar vortex (south of $\phi_{eq}$=70ºS), with larger values than the ones at the surf zone ($\phi_{eq}$=50ºS-30ºS). This behavior does not seem physical, and does not show up in ERA-Interim (Fig. R1b).

All of it evidences a rather nonphysical behavior of $\kappa_{eff}$ in WACCM in specific parts of the stratosphere, but is it a limitation of the effective diffusivity itself, or is there something in the PV field in WACCM that makes it unsuitable to be used as a well-behaved tracer for these calculations? A way to check would be to calculate $\kappa_{eff}$ using a diffusion-advection model to evolve a passive tracer (as in Abalos et al 2016, but for WACCM), and compare the results with $\kappa_{eff}$ calculated using PV. It is just a suspicion, but since the LCS-based diagnostic has been calculated using the wind field and it does not show the high mixing band in the summer stratosphere, it would not be surprising  that $\kappa_{eff}$ calculated using the same winds does not show it either. I am not asking the authors to perform those new $\kappa_{eff}$ calculations, which are computationally costly, but I suggest to edit the discussion on this issue to take these comments into account, specifically about the equivalent latitude coordinates. All in all, the absence of that nonphysical behavior in the other two mixing diagnostics speaks well of both of them.

[Figure]

Fig. R1: Mean effective diffusivity in JJA as a function of equivalent latitude and potential temperature, for a) WACCM (60-year mean), and b) ERA-Interim (34 year-mean). Both metrics have been calculated using potential vorticity fields on isentropes.

Other comments:

- Why is the analysis performed using a climate model output, and not a reanalysis? Are there any limitations (expected errors) because of the use of daily mean velocity fields for the trajectory calculations? Why using daily mean fields instead of instantaneous output?

- Calculation of the Lagragian Descriptor: What time interval [-tau, tau] has been used for the calculation? Is there a strong sensitivity of this parameter? The radiative timescales are much shorter in

the upper than in the lower stratosphere, so the consideration of PV as a passive tracer is a better approximation in the lower than in the upper stratosphere.

- Line 117: What is the "corresponding time interval" used to calculate the PDF of the norm of the gradient of M?

- Lines 120: LCS is signaling chaotic advection, not diffusion. Although both should be related.

- Fig. 2. I would suggest to plot the zonal mean zonal wind (in LCS-density and Deff) to easily relate the transport barriers and high mixing with the presence of zonal-mean jets.

- Fig. 2. Also, why showing only up to 60º latitude / equivalent latitude? It would be interesting to analyze mixing inside the vortex.

- Line 193. core of the polar vortices → polar jets .

- Fig. 3: Would it be useful to simply show the seasonal evolving average of the few years analyzed? It is difficult to discern summer or winter in this figure.

- Lines 324-325. On the relation between LCS and diffusion, which also appears in other parts of the text. Please correct if I am wrong, but LCS are defined for Hamiltonian (conservative) systems, so the trajectories performed are purely advective. What would be the tentative approach to quantitatively reconcile this paradigm with the notion of diffusion, which is an intrinsically non-conservative process?

Alvaro de la Cámara ---

---

## Author Comment (AC1)

Thank you very much for your detailed comments and suggestions, which have greatly helped improve the article. Below, we detail our responses to each of the referee points.

**1. Referee 1**

Main comments:

(1) Although stated in several parts of the text (lines 76-77, 314), it is not clear "what new perspective the new method provides on stratospheric mixing", since the comparison with other mixing diagnostics gives an overall agreement. I think the paper does not explore one obvious strength of the LCS-density diagnostic: the fact that it can diagnose large-scale stirring in longitude-latitude. The paper would greatly benefit from analyzing maps of LCS-density and relate the diagnostic with the main features of the stratospheric circulation, such as the location of the polar vortices, jets, regions of wave breaking, etc.

Thank you very much for the suggestion. A new section on this topic has been added to the manuscript which includes a new figure with longitude-latitude sections of the LCS density at various isentropic levels for both DJF and JJA 2007-2009. As an example, we include here the JJA case at 450K, where we can see differences in longitudes, showing the potential of this technique for comparisons with the main features of the stratospheric circulation, as the referee suggests.

[Figure]

(2) About section 3.2.1. If I understand correctly, the idea is that since equivalent latitude has a seasonal cycle in PV-coordinates, then the seasonal cycle of effective diffusivity ($\kappa_{eff}$) is affected by this feature and might be "contaminating" the physical significance of $\kappa_{eff}$. Latitude and equivalent latitude are not equivalent indeed, as stated in lines 242-243. The former is a geographical coordinate, the latter is a tracer contour-based coordinate: it represents the latitude at the edge of a polar cap enclosing the same area as a given tracer contour. Or in other words, it represents the area enclosed by a given tracer contour. Given a "well-behaved" tracer, at a given time there should be a one-to-one correspondence between a tracer contour value and $\phi_{eq}$. So if I am not mistaken, plotting the time-evolving PV as a function of $\phi_{eq}$ should be equivalent to plotting the time-evolving $\phi_{eq}$ as a function of PV (as in Fig. 5), it is a matter of remapping one as a function of the other. All this to say that the seasonal cycle of $\phi_{eq}$ in PV-coordinates should be the same as the seasonal cycle of PV in $\phi_{eq}$- coordinates. In fact, since the zonal-mean PV is monotic in latitude, one could perform a similar exercise and obtain a variable called "latitude" as a function of a "zonal-mean PV-coordinate"; one would expect an equivalent seasonal cycle in PVzonalmean(lat) and in lat(PVzonalmean). The point I am trying to make is that $\kappa_{eff}$, as long as it is computed using PV as a tracer, will indeed be affected by the seasonal cycle of PV (lines 257-260), but this should be independent of $\kappa_{eff}$ being defined in equivalent latitude coordinates, since PV has a seasonality in any coordinate. [By the way, it is surprising that the negative contour of PV at 800K in the last panel of Fig. 5 is placed at $\phi_{eq}$ 40ºN, it does not look consistent with the fourth panel of Fig. 5 where the $\phi_{eq}$=40ºN-contour is located at positive PV values.]

This is an interesting perspective, and we now discuss this in the manuscript. We did focus too narrowly on equivalent latitude instead of a more general discussion of tracer contour areas as being fundamental to the calculation of effective diffusivity and these not necessarily having a constant conversion to latitude depending on the tracer used. You are correct, and another tracer used in place of PV can be used define effective diffusivity or equivalent latitude. Then the seasonal cycle would be appear different if the tracer did not have the same seasonal cycle as PV. We have also changed the last panel of figure 5 due to the error you pointed out.

(3) This takes me to my next comment. That band of high $\kappa_{eff}$ at $\phi_{eq} = 20º$-40ºN in JJA (Fig. 6b) is indeed quite interesting. As reflected in the text, there should be no dynamical reason for high mixing to occur in that region since Rossby waves are filtered out below in the summer stratosphere. In fact, that band does not appear in the reanalysis ERA-Interim (see Fig. 1c in Abalos et al. 2016).

However, the $\kappa_{eff}$ calculations in Abalos et al. were performed by integrating a diffusion-advection model to evolve a passive tracer, as conceived by Nakamura (1996). So the question remains whether that band appears when using PV to calculate $\kappa_{eff}$. I happen to have calculated $\kappa_{eff}$ for the same WACCM runs (CCMI refc1) that are used in the present study, as well as for ERA-Interim, that I used in a study some years ago. I have plotted in Fig. R1 a similar plot as Fig. 6b, but for a 60-year mean in WACCM (Fig. R1a), and for a 34-year mean in ERA- Interim (Fig. R1b). I also get that high $\kappa_{eff}$ over 20º-40ºN in WACCM in a 60-year average, but that region is not present in the reanalysis, either computing $\kappa_{eff}$ with PV or with a passive tracer (see Fig. 1c in Abalos et al. 2016). This suggests that the high $\kappa_{eff}$ in the summer stratosphere is not a characteristic feature of $\kappa_{eff}$ computed with PV, but it is true that it does show up in WACCM. Moreover, Fig. R1a (WACCM) also displays a region with very high $\kappa_{eff}$ right inside the austral polar vortex (south of $\phi_{eq}$=70ºS), with larger values than the ones at the surf zone ($\phi_{eq}$=50ºS-30ºS). This behavior does not seem physical, and does not show up in ERA-Interim (Fig. R1b). All of it evidences a rather nonphysical behavior of $\kappa_{eff}$ in WACCM in specific parts of the stratosphere, but is it a limitation of the effective diffusivity itself, or is there something in the PV field in WACCM that makes it unsuitable to be used as a well-behaved tracer for these calculations? A way to check would be to calculate $\kappa_{eff}$ using a diffusion-advection model to evolve a passive tracer (as in Abalos et al 2016, but for WACCM), and compare the results with $\kappa_{eff}$ calculated using PV. It is just a suspicion, but since the LCS-based diagnostic has been calculated using the wind field and it does not show the high mixing band in the summer stratosphere, it would not be surprising that $\kappa_{eff}$ calculated using the same winds does not show it either. I am not asking the authors to perform those new $\kappa_{eff}$ calculations, which are computationally costly, but I suggest to edit the discussion on this issue to take these comments into account, specifically about the equivalent latitude coordinates. All in all, the absence of that nonphysical behavior in the other two mixing diagnostics speaks well of both of them.

Thank you for your detailed and thoughtful comment! We have added a discussion about these points in the conclusions of the new version of the manuscript. Specifically:

We acknowledge the potential discrepancies between $\kappa_{eff}$ calculations using PV and those using a passive tracer, as highlighted by the differences between WACCM and ERA data. We discuss the potential implications of the nonphysical behavior of $\kappa_{eff}$ in WACCM in certain regions of the stratosphere, as you have outlined. We also highlight the importance of using multiple diagnostics to assess stratospheric mixing, noting that the absence of nonphysical behavior in the other two mixing diagnostics used in our study supports their reliability.

Your suggestion to calculate $\kappa_{eff}$ using a diffusion-advection model to evolve a passive tracer in WACCM is excellent and could certainly help clarify whether the discrepancies arise from the use of PV or from model-specific biases. While we agree that performing these additional calculations would be ideal for resolving these uncertainties, we also recognize that this falls beyond the scope of our current study due to the significant computational cost. The fact that the age-based diagnostic does not show the high values suggests that a passive tracer would be better behaved also.

Other comments:

- Why is the analysis performed using a climate model output, and not a reanalysis? Are there any limitations (expected errors) because of the use of daily mean velocity fields for the trajectory calculations? Why using daily mean fields instead of instantaneous output?

  We use WACCM instead of reanalysis data for comparison purposes. Specifically, to apply isentropic eddy diffusivity, we need a high-resolution, spatially resolved Age of Air (AoA) field, which this particular WACCM run provides. Such a product is actually quite unusual, and reanalysis products do not have this. In addition, WACCM will have internally consistent physics.

  Regarding the use of daily mean versus instantaneous velocity fields, our initial choice of daily means was primarily due to computational time constraints. Since we were conducting a climatological study, we assumed there would be minimal differences in the results. However, as we repeated the simulations to include the polar regions, we now use instantaneous data every 3 hours, which offers better precision in trajectory calculations. As expected, this did not significantly change the results.

- Calculation of the Lagragian Descriptor: What time interval [-tau, tau] has been used for the calculation? Is there a strong sensitivity of this parameter? The radiative timescales are much shorter in the upper than in the lower stratosphere, so the consideration of PV as a passive tracer is a better approximation in the lower than in the upper stratosphere.

The referee is correct in this comment, and we appreciate the observation. We agree that using a $\tau$ parameter that varies with the constant potential temperature level would be more accurate, as radiative timescales indeed differ between the upper and lower stratosphere. However, this approach would significantly complicate the calculations. For this initial proof of concept of our method, we opted to use a const $\tau$ throughout the stratosphere, considering it a reasonable approximation. In this study, we set $\tau = 15$ days and focus on levels above 400K, where the effect of varying timescales is less pronounced. Future studies will explore this aspect in more detail.

Here is a small example of the LCS-density for a one-month period (January 2008) using different $\tau$ at 600K.

[Figure]

- Line 117: What is the "corresponding time interval" used to calculate the PDF of the norm of the gradient of M?
   The entire study period, i.e., 2007–2009, or the corresponding seasons or subsets thereof detailed in the article.
- Lines 120: LCS is signaling chaotic advection, not diffusion. Although both should be related.
   Fig. 2. I would suggest to plot the zonal mean zonal wind (in LCS-density and Deff) to easily relate the transport barriers and high mixing with the presence of zonal-mean jets.
   Fig. 2. Also, why showing only up to 60º latitude / equivalent latitude? It would be interesting to analyze mixing inside the vortex.
   Line 193. core of the polar vortices → polar jets.
   Fig. 3: Would it be useful to simply show the seasonal evolving average of the few years analyzed? It is difficult to discern summer or winter in this figure.
   All these changes and improvements have been implemented in the new version. Thank you very much for bringing them to our attention.
- Lines 324-325. On the relation between LCS and diffusion, which also appears in other parts of the text. Please correct if I am wrong, but LCS are defined for Hamiltonian (conservative) systems, so the trajectories performed are purely advective. What would be the tentative approach to quantitatively reconcile this paradigm with the notion of diffusion, which is an intrinsically non-conservative process?

You are correct that LCS are typically defined within the context of Hamiltonian (conservative) systems, where the dynamics are purely advective. However, experiments and observations suggest that transient-time diffusion can occur along LCS (e.g Lehahn et al. (2007); Tang and Walker (2012)). This has been discussed in detail in the new version

**2. REFEREE 2**

This work revisits the mixing in the stratosphere using on a metric based on the gradient of the so-called M-function which has already been used in several previous studies by the authors to visualize the Lagrangian coherent structures (LCS) in the flow. More precisely the module of the gradient of M is a metric for the local shear/strain along a Lagrangian trajectory and is akin to a measure of the horizontal isentropic mixing due to this shear/strain. This metric is here compared to effective diffusivity and another metric based on the age of air. This work is interesting and should eventually be publishable but it suffers from several issues in the present form that call for a major revision.

The main issue is a sort of reversal of the concept of proof. From an over simplified and a priori concept of what should be the mixing in the stratosphere, the results provided by the standard effective diffusivity are criticized to the benefit of the new metric. This cannot be a valid method in a system as complex as the atmosphere. The basic concept that less waves equal less mixing during summer is not necessarily entirely

true. First it can be that summer waves are less intense but more efficient at mixing as stated in d'Ovidio et al., 2009, and Shuckburgh et al., 2009. Technically it is related to the fact that stable and instable manifold are generating more mixing when they intersect at larger angle. Then during summer period in the northen hemisphere, the permanent Asian Monsoon anticyclone is a sort of turnstile that generates a large amount of mixing between subtropical and mid-latitude lower stratosphere with southward transport on the east side and northward transport on the west side. There is a quite abundant literature on the topic (see eg., Dethof et al., QJRMS, 1999, doi: 10.1002/qj.1999.49712555602 and Ploeger et al., JGR, 2013, doi: 10.1002/jgrd.50636). Other monsoons have similar albeit smaller effects. Therefore, it is not implausible that such reasons and others make summer stratospheric mixing more efficient than a naive expectation based on wave activity alone. In any case, the validity of a new measure of mixing should be proven on a controlled case and not claimed from weakly funded a priori expectations.

We appreciate your feedback and understand the concerns raised. Our intention was not to criticize effective diffusivity, which we acknowledge as a useful method for characterizing isentropic mixing properties in a zonal mean. In fact, we consider the work of Abalos et al. (2016) as a key reference in our analysis. We regret that this point may not have been clearly conveyed. We do believe that upper stratospheric summertime mixing should be small, and as reviewer 1 points out, different results can be obtained for calculating effective diffusivity by using a passive tracer rather than PV. With other metrics, this elevated summertime mixing does not exist in the upper stratosphere. In addition, we now visualize the effect of the monsoon directly by looking at the latitude-longitude maps of the LCS-density. We have added a more nuanced discussion of calculating effective diffusivity and the use of different tracers to do so.

Our study serves as a first proof of concept rather than as a "proof". While we have primarily focused on climatological zonal means for comparative purposes, we acknowledge the need for further testing to validate the new metric comprehensively. Future work will involve additional tests and analyses to better understand its performance and applicability. We have also incorporated a discussion on these considerations and relevant literature into the revised manuscript.

There are actually a number of good reasons to use equivalent latitudes in the stratosphere, especially for the polar regions (see e.g. Allen & Nakamura, 2003) and this cannot be discarded for weak reasons. I tend to think of effective diffusivity as an optimal method to characterize isentropic mixing properties in a zonal mean. The Gupta's method based on the age of air, although largely heuristic, takes into account vertical mixing as well. There is a need to characterize longitudinal variations of mixing, not manageable by effective diffusivity and this is where LCS based methods have some room.

We agree that equivalent latitudes are indeed invaluable for characterizing stratospheric dynamics. This is particularly true for comparing different in situ observations within the same season but at different times or different years. The seasonal cycle of of PV is not what is typically meant by mixing, however, and that seasonal cycle is showing up in the mixing diagnostic. We have included a more detailed discussion on the relevance and advantages of equivalent latitude in the revised manuscript. This discussion emphasizes the complementary roles of the three methods used and how they address different aspects of mixing. Additionally, we have added a new section on the longitudinal variations of mixing.

Besides this issue, the manuscript is often lacking of precision on many points. It is for instance very difficult to understand what is really shown as the LCS metric. L. 115-116 mention a maximum of the normalized gradient of M. If it is normalized and how, what is the meaning of the maximum? Then it is said that a PDf is calculated at each latitude, longitude and altitude but nothing is mentioned on the range of elements that enter this PDF nor of its normalization (density of PDF?). Then figure 2 is said to show the zonal average of this PDF but an averaged PDF is still a PDF not a single value. There are several possible ways to correct these statements so that they make sense but this is not the job of the reviewer. This is not helped by the fact that in general figures show quantities without unit and figure 1 lacks even a scale. In several instances of the manuscript the so-called density of LCS is quantitively compared to the effective diffusivity but we do not have the smallest idea of what should be the functional relation between these two quantities. There is no reason to expect a simple linear law.

Thank you for your detailed feedback. We apologize for the lack of clarity and have made several revisions to address these issues.

Other more minor points

- The function M and its gradient is not really a Lagrangian invariant as it would be modified by an arbitrary rotation on the sphere. This is probably of minor concern here but one should not claim a mathematical property which is not satisfied.

  We have modified the corresponding sentence.

- It is not possible to see the barrier effect of the polar jets in graphs that range from 60°S to 60°N in latitude.

  Thanks for the suggestion, we have extended the figures to the pole in the new version.

- The WACCAM model is used here at very low resolution and there is no surprise in the fact that the PV estimate is noisy. PV is also very badly conserved by the dynamics in such configuration. Such resolution is usually imposed by a whole set of chemical components and complex chemistry but here only velocities are used. It is hard to understand the rational of this choice when reanalysis at much higher resolution and with good PV conservation properties are available.

  We originally used WACCM for comparison purposes because the isentropic eddy diffusivity metric requires a high spatial and temporal resolution AoA dataset, and such a product is not generally available and certainly not from reanalysis (standard AoA output from models is zonal mean monthly mean, and AoA is not a standard output from reanalysis). The reviewer's point here, however, is actually quite helpful. This gets at another motivation for using something besides effective diffusivity (especially calculated using PV!) as a metric for mixing. When we compare the BDC strength across models and reanalysis, we can use the TEM vertical velocity equally well for both. If having output from a coarse resolution model precludes accurate calculations of mixing not because the model is not representing mixing well but because the method requires something better, then we need a different method. LCS density is not the only possible answer to this problem, but because of the properties of its Lagrangian approach, it can be used with a wide range of resolutions (see Badza et al. (2023) for more on LCS and uncertainties in data). When we used higher temporal resolution when we re-calculated the metric for these revisions, for example, we found very little difference from using daily.

- The limitations of the model are perhaps a reason of the discrepancies observed here. The differences between the diagnosed effective diffusivity shown in fig. 2 and that shown in Abalos et al., 2016 are quite significant and impact on the discussion.

  See above response. As a community, we need mixing metrics that are useful for models at a variety of resolutions as well as reanalysis products.

- The text on l. 215-217 does not match what I see on figure 3. There is a strong seasonal cycle in the effective diffusivity in particular at 800K. It is difficult to appreciate its amplitude at 450K due to the saturated color bar. It looks smaller than for the LCS density but again, how can the two quantities be compared?

  Figure 3 and the corresponding text have been modified to more accurately reflect the observations regarding the seasonal cycle in effective diffusivity.

- Some hints on the sensitivity of the LCS density to the time parameter tau should be mentioned.

  We have added a brief discussion on the sensitivity of the LCS density to the time parameter $\tau$. See also the answers to the referee 1

- The whole discussion starting on line 246 seems to me misleading and useless. There is no mystery that the effective diffusivity is large where the relation between equivalent latitude and PV (or tracer) fluctuates. This is in some way a built-in property of the method. There is no problem either in the fact that the relation between equivalent latitude and PV displays a seasonal cycle. In any case, the area enclosed within an equivalent latitude is the same as the area enclosed in this latitude. In passing, I m a but surprised that a Q contour with negative value is located at 40°N.

  We have revised the discussion and corrected the error you pointed out.

- The paragraphs starting on l. 107 and 115 are somewhat repetitive.

  Thanks. We have changed the sentence.

**References**

Abalos, M., Legras, B., and Shuckburgh, E.: Interannual Variability in Effective Diffusivity in the Upper Troposphere/Lower Stratosphere from Reanalysis Data, Quarterly Journal of the Royal Meteorological Society, 142, 1847–1861, https://doi.org/10.1002/qj.2779, 2016.

Badza, A., Mattner, T. W., and Balasuriya, S.: How sensitive are Lagrangian coherent structures to uncertainties in data?, Physica D: Nonlinear Phenomena, 444, 133 580, 2023.

Lehahn, Y., d'Ovidio, F., Lévy, M., and Heifetz, E.: Stirring of the northeast Atlantic spring bloom: A Lagrangian analysis based on multisatellite data, Journal of Geophysical Research: Oceans, 112, https://doi.org/https://doi.org/10.1029/2006JC003927, 2007.

Tang, W. and Walker, P.: Finite-time statistics of scalar diffusion in Lagrangian coherent structures, Phys. Rev. E, 86, 045 201, https://doi.org/10.1103/PhysRevE.86.045201, 2012.

---

## Author Response (AR2)

Dear Editor Aurelien Podglajen,

Thank you for your message and for considering our manuscript "Lagrangian Coherent Structures to Examine Mixing in the Stratosphere". We truly appreciate the referees' thoughtful review and constructive feedback. In response to Referee 1's comment, we have expanded the discussion on the limitations of using PV to compute effective diffusivity in the conclusion and have also added a clarifying sentence in the equivalent latitude section. The changes are highlighted in blue in the track-changes file.

We are grateful for your time and consideration and look forward to your opinion.

Best regards,

Marianna and Jezabel